# Characterization of Disseminated Tumor Cells (DTCs) in Patients with Triple-Negative Breast Cancer (TNBC)

**DOI:** 10.3390/cells14120857

**Published:** 2025-06-06

**Authors:** Anne Eckardt, Ivonne Nel, Laura Weydandt, Elisa Brochwitz, Anne Kathrin Höhn, Karsten Winter, Bahriye Aktas

**Affiliations:** 1Department of Gynecology, Medical Center, Leipzig University, 04103 Leipzig, Germany; 2Department of Pathology, Medical Center, Leipzig University, 04103 Leipzig, Germany; 3Institute of Anatomy, Medical Faculty, Leipzig University, 04103 Leipzig, Germany

**Keywords:** triple negative, breast cancer (TNBC), disseminated tumor cells (DTCs), bone marrow, neoadjuvant chemotherapy, pathologic complete remission (pCR), Ki67, HER2, PD-L1

## Abstract

Triple negative breast cancer (TNBC) is the most aggressive molecular subtype and it lacks targetable receptors. Patients have an increased risk of recurrence and poor prognosis. Little is known concerning the characteristics of disseminated tumor cells (DTCs) and their role in TNBC patients. We analyzed the bone marrow aspirates of 80 patients with primary (*n* = 67) or recurrent (*n* = 13) TNBC, using a multi-parameter immunofluorescence staining procedure, including Pan-CK as an epithelial marker, vimentin (vim) as a marker of epithelial–mesenchymal transition, Ki67 for cell proliferation, and HER2 as well as PD-L1 as therapy-related markers. The DTC positive rate was 56% (*n*= 45) among the cohort. We found 20 different DTC subpopulations. The most frequently detected profile was CK+Vim+Ki67+ (*n* = 75 cells). The occurrence of CK- DTCs (*n* = 69) was significantly correlated to PD-L1 (r = −0.305, *p* < 0.01) and HER2 positivity (r = −0.234, *p* < 0.001). DTC positive patients that received neoadjuvant chemotherapy (NACT) and did not reach pathologic complete response were more likely to have CK- DTCs. Our data indicate that the occurrence of DTC subpopulations positive for Vim, Ki67, and HER2 appear to be markers for bad prognosis and could be therapeutically relevant. Furthermore, our results raise the question of whether DTCs are dormant in TNBC patients and persistent towards chemotherapy.

## 1. Introduction

Although therapy options for breast cancer patients have improved, about 20% suffer from recurrence within 10 years after primary treatment [1]. This could be caused by disseminated tumor cells (DTCs), which dissociate from the primary tumor as a very early metastatic event during the disease course. Through lymph and blood vessels they can befall distant sites such as the bone marrow where they enter a latent state [2]. For reasons which are not yet entirely known, DTCs can re-awake after many years, re-circulate, and they may cause local or distant relapse. The prognostic value of DTCs in patients with early breast cancer and the association with declined survival has been shown in numerous studies [3,4,5,6].

Breast cancer treatment is challenged by the enormous heterogeneity of the tumor cells, which are prone to undergoing phenotypic and genetic alterations during disease progression and under therapy leading to discordant features of DTCs compared to the primary tumor [7,8,9,10]. However, tumor properties and hence treatment decisions are assessed at diagnosis and are based on an initial tissue biopsy. According to the St. Gallen consensus, breast tumors can be divided into defined molecular subtypes [11].

The most aggressive subtype is triple negative breast cancer (TNBC), which lacks expression of the hormone receptors estrogen (ER)- and progesterone (PR)- receptor as well as the human epidermal growth factor receptor 2 (HER2) [12,13,14]. The expression of HER2 is usually determined by immunohistochemical staining (IHC) of the tumor tissue based on the current guidelines with exclusively circular membranous staining. Depending on the strength and pattern of the staining, a score is used to evaluate the HER2 expression. Tumors that scored 0 (no staining) are defined as HER2 negative, and specimens evaluated with 1+ (weak staining in ≤10% of tumor cells) are classified as HER2 low. Cases that scored 2+ (moderate staining in >10% of tumor cells) are subjected to an additional (fluorescence/chromogen) in situ hybridization (FISH or CISH) assay to determine the HER2 gene amplification levels at the chromosomal level [15,16,17,18]. If FISH/CISH remains negative, the 2+ tumor is defined as HER2 low whereas FISH/CISH positivity (amplification) classifies a 2+ tumor as HER2 positive [18]. Specimens that scored 3+ (strong staining in >10% of tumor cells) mean HER2 positive tumors. The delicate classification of HER2 expression levels is relevant for targeted treatment options directed against HER2.

A tumor is diagnosed as TNBC if <1% of the tumor cells express ER and PR, and HER2 is scored 0 or 1+ or 2+ /FISH/CISH negative (HER2 low), while the proliferation index is usually high (Ki67 >25%) [19]. Due to the lacking targets and aggressive tumor biology, patients commonly receive neoadjuvant chemotherapy (NACT) but have an increased risk of recurrence within five years and poor prognosis. Particularly, patients who do not achieve pathological complete remission (pCR) after NACT may require targeted treatment approaches. Several trials in these patient populations are currently ongoing.

The Phase III Destiny-Breast04 trial showed survival benefit of patients with HER2 low metastatic breast cancers, including a small number of TNBC populations, treated with the antibody drug conjugate trastuzumab deruxtecan (T-DXd) compared to chemotherapy [20]. Other important targets in cancer therapy, especially in immunooncology, are the programmed cell death-ligand 1 (PD-L1) and the programmed cell death protein (PD-1), which promote tumor cell-specific T cell inactivation or apoptosis and hence lead to tumor cell growth and enhance tumor immune escape [21]. PD-L1 is described as a marker of bad prognosis in solid tumors [22]. The use of immune checkpoint inhibitors such as the PD-1 inhibitor pembrolizumab became a promising therapeutic option for TNBC patients [23]. The combination of chemotherapy with pembrolizumab has led to a 14% increased pCR rate, a significantly improved prognosis and thus, was approved in 2022 [24].

For the time being, pCR after NACT is the most important prognostic factor for patients with TNBC [25]. Although DTCs are considered an independent prognostic factor for recurrences and distant metastases [3,4], little is known concerning the characteristics of DTCs and their role in TNBC patients. We assume the existence of DTC subpopulations that are able to resist systemic or targeted therapy and might pander to relapse and metastasis. In order to leave the primary tumor, the epithelial-to-mesenchymal transition (EMT) cells appear to be a key event linked with increased motility, metastatic potential, invasiveness, and chemo-resistance [26,27,28]. After performing EMT, the appearance and behavior of tumor cells might be discordant to the primary tumor. As a result, DTCs with not only epithelial, but EMT-like or mesenchymal properties might occur and could be missed by detection methods solely based on epithelial markers such as pan-cytokeratin (Pan-CK).

Previous methods for detecting these rare cells have relied on immunocytological staining of the epithelial marker cytokeratin (CK) [29]. However, recent research results have shown that particularly DTCs with mesenchymal properties are associated with a poorer prognosis [30,31,32,33]. Despite their potential as independent prognostic factors, no clinical test is available to assess patients’ risk for recurrence and no targeted treatment is obtainable based on their DTC profile at primary diagnosis. Adjuvant treatment with bisphosphonates has been shown to eliminate epithelial DTCs in the bone marrow of breast cancer patients [34], but targeted treatment would be desirable. Therefore, the aim of this study was the detection and characterization of epithelial and non-epithelial DTCs by sequential multi-parameter immunofluorescence staining and to investigate their potentially prognostic value in patients with TNBC.

## 2. Materials and Methods

### 2.1. Study Population and Informed Consent

In this study, we enrolled patients who were treated at the Department of Gynecology at the University Hospital Leipzig, Germany, between 2019 and 2023. Bone marrow aspirates from patients with histopathologically confirmed TNBC were sampled during surgery after they had agreed to participate in the trial and signed a written informed consent in accordance with the requirements of local ethics committee (internal reference number: No. 216/18-ek), which is a standard procedure in our trials. Like in previous studies [30,35], we collected clinicopathological patient data from the medical records, and the patient characteristics are summarized in Table 1.

In this study, we included a total of 80 patients. According to clinical routine, core needle biopsies (CNB) of the breast tumor were obtained at diagnosis. As our study focused on TNBC, all patients were negative for estrogen receptor (ER) and progesterone receptor (PR). Additionally, 64 patients were negative (IHC DAKO Score 0) for the human epidermal growth factor receptor 2 (HER2) and 16 patients were “HER2 low” (*n* = 12 IHC Score 1+; *n* = 4 IHC Score 2+, CISH negative) [11,18]. The median age was 50 years, ranging from 26 to 86 years at the time of diagnosis. In the cohort, 83% (*n* = 67) of the patients presented with primary and 16% (*n* = 13) had recurrent breast cancer.

### 2.2. Bone Marrow Aspirates

During surgical tumor resection bone marrow aspirates were collected from both sides of the anterior iliac crest and prepared in our laboratory as described previously [30]. Shortly, after density gradient centrifugation, cell suspensions were transferred onto glass slides (1 × 10^6^ cells per slide) using a cytospin centrifuge (Tharmac, Limburg a.d. Lahn, Germany), fixed with ice-cold methanol and stored at 4 °C until subjected to immunocytochemical staining. We prepared 8 slides per bone marrow aspirate. Remaining bone marrow cell suspensions were stored in liquid nitrogen for further investigation [30,35]. In this study, bone marrow aspirates of 80 patients who met the criteria for a diagnosed TNBC were included.

In parallel, we analyzed cytokeratin-positive (CK+) DTCs in the same bone marrow aspirate (4 of 8 slides) following a concept for standardized DTC detection using brightfield microscopy [29] with an antibody directed against Pan-CK and labeled with alkaline phosphatase (Cat. No.130-090-462, Miltenyi Biotech, Bergisch Gladbach, Germany), based on the optimized protocol established in our laboratory (Figure 1) and published elsewhere [30,35]. In brief, DTCs were visualized in pink using alkaline phosphatase and short counterstaining with hematoxylin, which colored the nuclei light blue. DTCs were semi-automatically detected and enumerated using the Aperio Versa microscope-based scanning system (Leica Biosystems, Nußloch, Germany) with rare events software that was trained to select DTC candidates according to color, shape, intensity, and size [30,35]. Two independent investigators, a certified cytologist and a trained pathologist, individually evaluated cell morphology and cytological staining patterns of the selected DTC candidates. Hence, DTCs of each included bone marrow aspirate were quantified with both a standard brightfield method [30,35] and the novel sequential multiparameter staining described in the following paragraph.

### 2.3. Cell Culture for Reference Slides

As control cells, similar to our previous studies [30,35], we used cells from the hormone-receptor-positive, HER2 low breast cancer cell line ZR75-1, the MCF-7 breast cancer cell line, the fibroblast-like glioblastoma cell line T98G, and the epithelial–mesenchymal cervical carcinoma cell line CaSki. The cells were initially obtained from the American Tissue Culture Collection (ATCC). Cryo-preserved stock solutions were thawed and maintained under standard conditions (37 °C and 5% CO_2_ atmosphere) in either Dulbecco’s Modified Eagle Medium (DMEM) or Roswell Park Memorial Institute (RPMI)-Medium (CaSki) containing 4.5 g/L glucose and l-glutamine (Cat. No. FG 0435, Biochrom, Berlin, Germany) supplemented with 10% fetal bovine serum (Cat. No. S 0615, Biochrom, Berlin, Germany) and 100 U/mL penicillin/streptomycin. After the cells were harvested with trypsin/EDTA solution, they were re-suspended in media and left in the incubator for 15 min to allow cell surface recovery. After washing in phosphate-buffered saline (PBS), they were counted using the Neubauer chamber. Then 5 × 10^5^ bone marrow cells were spiked with 25.000 ZR75-1, T98G and CaSki cells, respectively, resulting in 20:1 ratio for each cell line. MCF-7 cells were used separately in a concentration of 200.000 cells per ml PBS. Cells were spun onto slides using a cytospin centrifuge before fixation with ice-cold methanol for 5 min [30,35].

### 2.4. Sequential Immunofluorescence Staining and DTC Imaging

We modified a previously established sequential multi-parameter imaging method [30,35] based on immunofluorescence (IF) staining that allowed the investigation of Pan-CK as an epithelial marker, vimentin (Vim) as a marker of epithelial–mesenchymal transition, Ki67 for cell proliferation, and HER2 as well as PD-L1 as therapy-related markers on the same DTC. Counterstaining against the common leucocyte antigen CD45 was used to exclude hematopoietic cells and DNA in cell nuclei was stained using DAPI.

Per patient, 2 × 10^6^ prepared bone marrow cells were subjected to a sequential multi-parameter staining procedure applying antibodies with releasable fluorochrome conjugates. The Axioscan 7 scanning microscope (Carl Zeiss Microscopy, Jena, Germany) and Zeiss Zen 3.8 software allowed annotation and localization of single cells. The use of releasing enzymes made it possible to stain the same DTC multiple times against six different markers in total.

In brief, antibody-cocktail panel 1 (Pan-CK-APC REAdye_lease^TM^, Cat. No. 130-123-091; vimentin-FITC REAdye_lease^TM^, Cat. No. 130-127-022; CD45-PE REAfinity^TM^, Cat. No. 130-110-632, all by Miltenyi Biotec, Bergisch Gladbach, Germany) was applied to stain three markers simultaneously before the samples were covered with a mounting medium containing DAPI for DNA staining in the nucleus. Then, images of the DTCs were acquired.

To detect the secondary fluorescence, we collected microscopic scans of the stained slides using band pass excitation filters AF488 for FITC (emission 517 nm), Cy3 for phycoerythrin (PE; emission 561 nm), Cy5 for allophycocyanin (APC; emission 673 nm), and D for DAPI (emission 465 nm) at 20× magnification. After uncovering the slides, antibody–fluorochrome complexes APC and FITC were released from the cells using the REAlease Release Reagent (Cat. No. 130-120-675, Miltenyi Biotech, Bergisch Gladbach, Germany) to enable further immunofluorescence staining using panel 2.

In the second round of staining, antibody panel 2 (PD-L1-AF647, Cat. No. ab205921, abcam, Cambridge, UK; Ki67-FITC, Cat. No. 130-117-691; HER2-APC-Vio770, Cat. No. 130-124-470, both Miltenyi Biotec, Bergisch Gladbach, Germany) was applied. Since staining against CD45 was performed using a non-releasable antibody within panel 1, the PE signal remained detectable after staining with panel 2 using the excitation filters mentioned above plus Cy7 for Allophycocyanin-Vio 770 (APC-Vio770, emission 773 nm).

For every patient sample, we acquired two separate whole slide images: one for panel 1 using FITC, Cy3, Cy5, and DAPI, and one for panel 2 using FITC, Cy3, Cy5, DAPI, and APC-Vio770. These images were aligned and overlaid using the Zeiss Zen software. This alignment enabled us to analyze and quantify each cell across all six fluorescent channels. By overlaying the images of individual cells, we were able to assess distinct cellular profiles based on the positivity or negativity for markers such as Pan-CK, vimentin, CD45, PD-L1, Ki67, and HER2. Cells that were CD45 negative but had a positive nuclear staining with DAPI and a positive staining against Pan-CK, Vim, HER2, Ki67, PD-L1, or multiple markers at once were defined as tumor cells.

As a positive and negative control with each run and in accordance with previous experimental set ups [30,35], we used reference slides with a mix of bone marrow cells and ZR75-1 (breast cancer cells), CaSki (cervical carcinoma cells) plus T98G (glioblastoma cells) as wells as slides with MCF-7 cells, which were prepared, stored, and fixed in the same manner.

### 2.5. Release of Fluorochromes from APC- and FITC- Conjugated Antibodies for Further Staining

After staining the first antibody panel, using releasable antibodies against Pan-CK and vimentin, the cells were treated with release reagent (Cat. No. 130-120-675, Miltenyi Biotech). Next, we analyzed the average intensity of fluorescent dye in cellular compartments in positive control cells before and after the release procedure as we have done in a previous study [30]. In this study, we calculated the signal strength based on integrated intensity of pixels and percentage coverage of the color within and across cellular compartments using Mathematica (Version 13, Wolfram Research Inc., Champaign, IL, USA). First, the regions to be analyzed were exported from the Zen software as RGB images, containing three color channels red (R), green (G), and blue (B), which were separated into individual channels. The images were subsequently loaded into Mathematica software and split again into the two individual channels red (APC) and green (FITC). The images were smoothed using an edge-preserving filter [36]. Cell detection was then performed using an automatic thresholding method based on Otsu’s algorithm [37], and specific cell masks were created. A morphological closing was applied to remove small segmentation artifacts (<1000 pixels), and morphological filling was used to close gaps in the cell masks. This process allowed the creation of red and green cell masks that identified cells in the respective channels before and after the release step. Within these masks, the average fluorescence intensity per cell was determined both pre- and post-release. The ratio of these values corresponded to the release rate in the red and green channels.

### 2.6. Immunohistochemical Staining of Matching Tumor Tissue

For our study, we performed re-analysis of HER2 and PD-L1 of DTC-positive patients using formalin-fixed, paraffin-embedded (FFPE) specimens from tumor tissue and if available from synchronous axillary lymph node metastases. As mentioned above, the material was obtained by core needle biopsy (CNB) at primary diagnosis or during primary oncological surgery. In the case of multiple positive lymph nodes occurring during surgery, we chose one specimen for immunohistochemical staining and evaluation. Immunohistochemical examination was performed in our Department of Pathology according to the ASCO/CAP guidelines using the Ventana-platform (Roche, West Sussex, UK) [16,38,39,40].

### 2.7. Statistical Analysis

We used Spearman Rho rank correlation and descriptive statistics to explore associations between DTC subpopulations and clinical characteristics of the patients. To identify correlations between cellular markers, we separated the occurring DTC profiles into various groups, such as CK positive and CK negative. Clinical data were divided according to characteristics such as neoadjuvant therapy and pCR. Further, we used Mann–Whitney U and Chi-Square test to analyze differences in the patients’ DTC profiles. The SPSS (IBM SPSS Statistics, Version 29.0.0.0 (241) program was used, and statistical significance was set at *p* < 0.05.

## 3. Results

### 3.1. Staining of Reference Slides Using Sequential Immunofluorescence

#### 3.1.1. Control Cells for Marker Panels

To investigate DTC subpopulations based on epithelial, mesenchymal, and therapeutically relevant markers in patients with TNBC, we modified our previously established multi-staining method [30], as described above (Figure 2). Thus, we used MCF-7 cells and the prepared mixes of cell suspensions from bone marrow spiked with ZR75-1 breast cancer cells (epithelial phenotype), T98G glioblastoma cells (mesenchymal phenotype) and CaSki cells (epithelial–mesenchymal phenotype) as positive and negative controls for each run.

#### 3.1.2. Calculation of the Release Rate

The release rate was calculated as described above in Section 2.5. In total, we analyzed 1179 of the ZR75-1 cells that stained positive for Pan-CK-APC (red) and 3106 of T89G cells that stained positive against Vim-FITC (green). Thus, we could confirm a 93.4% and 85.1% reduction in dye intensity in the APC (red) and FITC (green) channels, respectively (Figure 3 and Figure 4). Due to persisting autofluorescence in the bone marrow samples, signal reduction was lower in the green compared to the red channel. However, the background signal was also detected in the negative controls and accounted accordingly during analysis.

Hence, secondary fluorescence of Pan-CK- APC and Vim-FITC was successfully quenched, and another set of antibodies could be applied. In addition, the markers vimentin (used in the first panel) and Ki67 (used in the second panel), labeled with the green dye, showed different expression patterns.

#### 3.1.3. Characterization of Control Cells

After releasing the Pan-CK and vimentin signals from panel 1, cells were labeled with panel 2, namely antibodies against PD-L1, Ki67, and HER2. CD45 remained on the cells and DAPI was applied together with the mounting media after the second staining. Then, cell images were recorded, and tumor cells were detected and quantified according to the above-mentioned criteria. Next, we performed manual tumor cell profiling using the Zen software (Zeiss). As expected from previous experiments [30], in this set up the epithelial ZR75-1 breast cancer cells stained positive for Pan-CK, HER2 and Ki67, whereas the T98G glioblastoma cell line was positive for Ki67 and vimentin but remained negative for Pan-CK and HER2 (Figure 5 and Table 2).

The CaSki cells stained positive for vimentin, Ki67, Pan-CK, as well as PD-L1 but did not express HER2. MCF-7 cells were positive for Pan-CK and Ki67, but negative for vimentin, PD-L1, and HER2. The four cell lines were CD45 negative. Hematopoietic cells were negative for all the above-mentioned markers besides vimentin and partially PD-L1, displaying a positive nuclear staining using DAPI, though (Table 2).

### 3.2. Identification and Quantification of DTC-Subtypes in Bone Marrow Samples

For the investigation of patient-derived bone marrow samples, we defined tumor cells as cells exhibiting an increased nucleus/plasma-ratio. Further, tumor cells were negative for CD45, showed a positive nuclear staining with DAPI and a positive staining against either Pan-CK, Vim, HER2, Ki67, PD-L1, or a combination of the latter ones. Cells that met the above-mentioned criteria were recorded and counted as tumor cells. In contrast to immunohistochemical staining in tissue, and similar to previous work [30], we did not apply any scoring. Based on the fluorescent signal, a cell was either positive or negative for a marker such as Ki67, PD-L1, or HER2 in this study.

Figure 6 shows representative images of DTCs from patient samples exhibiting different profiles.

In total, we detected 253 DTCs using the sequential immunofluorescent staining procedure (Table A1). Based on the antibody panels used, we were able to find 20 different DTC subpopulations. The most frequently detected profile was CK+Vim+Ki67+ (*n* = 75 cells) followed by CK+Vim+PDL1+Ki67+ (*n* = 37 cells) and CK+ (*n* = 30 cells; Figure 7A).

Interestingly, we found CK negative (CK-) cells (*n* = 69) with the following predominantly occurring profiles: Vim+Ki67+ (*n* = 21), Vim+PDL1+Ki67+ (*n* = 17), and PDL1+/Ki67+ (*n* = 15, Figure 7B).

The applied marker panels 1 and 2 resulted in a matrix structure consisting of phenotypic subpopulations reflecting the process of EMT reaching from epithelial (CK+Vim-) to mixed (CK+Vim+) and mesenchymal (CK-Vim+) as well as stem-like features (CK-Vim-) combined with therapeutically relevant markers PD-L1, Ki67, and HER2 (Figure 8). Spearman rho correlation revealed that the occurrence of CK- DTCs was significantly correlated to PD-L1 (r = −0.305, *p* < 0.01) and HER2 positivity (r = −0.234, *p* < 0.001). Further, Ki67 positivity was significantly correlated to Vim+ (r = 0.435, *p* < 0.01) DTCs.

### 3.3. DTC Subtypes in Primary and Recurrent TNBC Patients

Among the cohort, 45 patients (56%) were DTC positive, with 38 being primary cases and seven being recurrent cases. There were no statistically significant differences in DTC status or DTC phenotypes between primary and recurrent cases. Descriptively, the mean DTC count was higher in recurrent TNBC patients compared to those with primary TNBC (6.7 vs. 5.4). Additionally, the mean of DTCs with a mixed phenotype (CK+Vim+) was increased in recurrent patients (4.1 vs. 2.9), as was the mean of CK+ DTCs (5.3 vs. 3.9), the mean of Vim+ DTCs (5.4 vs. 3.8), and the mean of Ki67+ DTCs (5.6 vs. 3.9). However, the mean of PD-L1+ DTCs was lower in recurrent cases compared to primary cases (0.7 vs. 2.6), and no HER2+ DTCs were detected in recurrent cases (0 vs. 0.55).

Further, we compared DTC profiles based on the menopausal status of the patients and found a decreased DTC count in peri-menopausal patients (*n* = 5, mean 4.8), along with decreased epithelial (CK+Vim-) DTCs (mean 0.2) and increased HER2+ DTCs (mean 1.2) compared to post- (*n* = 21) and pre- menopausal (*n* = 19) DTC positive patients. Interestingly, the mean of CK+ DTCs was higher in pre-menopausal patients compared to peri- and post-menopausal cases (4.4 vs. 3.8 vs. 3.9).

Looking at the tumor grading, we found an increased mean of HER2+ DTCs in patients with grade 3 (*n* = 28) vs. grade 2 (*n* = 15) tumors (mean 0.6 vs. 0.3) and increased mesenchymal DTCs (CK-Vim+) in grade 3 vs. 2 (mean 1.1 vs. 0.6) along with increased CK- cells (mean 1.7 vs. 1.1) in cases with grade 3 vs. grade 2 tumors.

### 3.4. DTC Status and Subtypes After NACT in Patients with Primary TNBC Reaching pCR vs. Non-pCR

We looked at the DTC status (positive/negative) of all TNBC patients with primary tumors who received NACT (*n* = 57) and found no significant difference between the pCR vs. non-pCR groups (x^2^ = 0.449, *p* = 0.503). The DTC mean was increased when patients did not reach pCR; the difference was not statistically significant, though (6.9 vs. 4.9, *p* = 0.164). We investigated DTC profiles of patients with primary tumors who received NACT (*n* = 32) and reached pCR (*n* = 19) vs. those with non-pCR (*n* = 13).

Chi square test revealed a statistically significant difference in DTC phenotypes between patients with pCR and non-pCR (x^2^ = 4.979, *p* = 0.026; Figure 9). Patients without pCR were more likely to display both CK+ and CK- DTCs compared to the pCR group.

Since the total DTC numbers varied among patients, we looked at the mean proportion of CK+ and CK- DTCs in the pCR vs. non-pCR group and found an increased ratio of CK- DTCs in patients without pCR (z = −1.561; *p* = 0.118). The mean of DTCs with mixed (CK+Vim+) and mesenchymal phenotypic characteristics (CK-Vim+) was increased (4.2 vs. 2.2 and 1.2 vs. 0.8) in the non-pCR group as well as mean values for CK+, CK-, Vim+, Ki67+ and HER2+ DTCs.

Mann–Whitney-U test revealed a tendency towards significance in the difference between CK-Vim+ DTCs in the pCR vs. non-pCR group (z = −1.707; *p* = 0.088) showing that the proportion of DTCs with mesenchymal features (CK-Vim+) was increased in patients that did not reach pCR. Chi square test showed that significantly more patients with non-pCR presented CK-Vim+ cells compared to those who reached pCR (x^2^ = 4.394, *p* = 0.036). The mean of solely epithelial DTCs (CK+Vim-) was decreased in patients without pCR (0.7 vs. 1.2).

### 3.5. DTC Subtype Detection vs. Standard Detection of CK+ Cells

Using the standard brightfield DTC detection method [35], we found that 37 of the 80 patients were CK+ DTC positive with a mean count of 6.3 DTCs per patient and 43 cases were DTC negative. Applying our sequential IF approach to bone marrow samples from the CK+ DTC positive cases, we confirmed DTC positivity in 33 cases with a mean DTC count of 6.3; however, four cases remained negative. Further, we subjected samples from the CK+ DTC negative patients to our sequential IF staining and found 12 positive cases among them with a mean count of 3.8 DTCs. Of these twelve positive cases, all had CK+ DTCs and six presented mesenchymal DTCs (CK-/Vim+) in addition.

### 3.6. Correlation of DTC Analysis to Clinical Parameters

We used Spearman rho correlation to investigate possible connections of DTC numbers to age, BMI, and Ki67 index in the core needle biopsy (CNB) gained at diagnosis without any significant results. The number of Ki67+ DTCs was not correlated to the Ki67 index in the tumor tissue.

Among the DTC positive patients, we found nine cases with HER2+ DTCs (ranging from one to six cells) and compared the HER2 expression in the matching tumor tissue either from the CNB or resected during surgery if no pCR was reached. We found a receptor concordance between DTCs and tissue in four cases: 2× CNB (Score 1+) and 2× resected tissue (Score 1+). Further, in four cases tissue from lymph node metastases were available and we found HER2 expression in two (Score 1+ and 3+) of the nine cases with HER2+ DTCs, whereas the matching tumor tissue in the latter one (Score 3+) was HER2 negative.

In 27 patients, we detected PD-L1+ DTCs (ranging from 1 to 15 cells) and compared PD-L1 expression in matching tumor tissues when available (*n* = 20). Concordance was found in 10 patients: 2× CNB and 8×resected tissue in cases without pCR. In nine patients, tissue specimens from lymph node metastases were available and three of them were positive for PD-L1 (Table 3).

## 4. Discussion

In this study, we investigated DTC subpopulations in a TNBC cohort using a sequential staining method with releasable antibody–fluorochrome-conjugates based on our previously reported multi-parameter analysis [30]. Similar to former reported technical challenges, the employment of releasing enzymes leads to fluorochrome quenching rates of 93.4% and 85.1%, respectively (previous study: 94.2% and 84.6%). With the aid of adequate control cells, sequential detection of the phenotypic cellular markers Pan-CK, vimentin and CD45 as well as the therapeutically relevant markers Ki67, HER2, and PD-L1 was feasible. We were able to identify distinct DTC profiles not solely based on epithelial features.

DTCs were defined as cells that had a nucleus (DAPI), were CD45 negative, and presented an increased nucleus/plasma ratio. Using the sequential staining approach, we detected DTCs in 56% of the patients (45/80). Having in mind that 80% of the patients (64/80) received NACT, the DTC positive rate is in line with results of a study by Volmer et al. who found that 45% of breast cancer patients were DTC positive after NACT using the CK-based standardized staining method [41]. In our study, the standard brightfield method revealed a positive DTC status in 46% of the patients (37/80). Using the multi-parameter DTC subtype detection method resulted in identical mean DTC quantities compared to the standard detection of CK+ cells. Four cases remained negative using the novel approach. A possible cause could be the use of different cytospins processed from the same sample. Since DTCs are rare events, they might be distributed patchily within the cell suspension and hence the slides. Interestingly, among the cases that were CK+ DTC negative according to the standard method we found an additional 12 positive cases using the multi-parameter approach.

We observed an increased proportion of CK- DTCs in patients who did not reach pCR. Particularly, mesenchymal CK-/Vim+ DTCs were elevated in non-pCR cases, while solely epithelial DTCs were decreased. Vice versa, patients who achieved pCR were likely to present only epithelial DTCs. It was already reported that chemotherapy might cause the occurrence of resistant DTCs with mesenchymal and stem cell-like features [31,33,42,43,44]. However, it has been shown that NACT can eradicate (epithelial) DTCs. A sub-analysis of the GeparX trial revealed that TNBC patients who were DTC positive before NACT (*n* = 7) had lower pCR rates compared to DTC negative patients (41.2% vs. 60.4%, *p* = 0.267). Further, they showed that TNBC patients with detectable (CK+) DTCs after NACT had a 25% decreased pCR rate (not statistically significant). This sub-study, however, took only CK+ DTCs into account that were determined using the standard brightfield method [45]. In our study, the majority of the patients with mesenchymal DTCs, which are associated with chemo-resistance, did not reach pCR, probably due to mesenchymal or EMT-like tumor cells in the primary tumor tissue. Concerning this assumption, further research is required.

Our findings indicate that not only the DTCs status or count, but the DTC phenotypes might be prognostic factors. Looking at the phenotype, we found 20 patients with solely epithelial DTCs (CK+), while 24 patients presented CK+ and CK- DTCs. In one case, we detected only CK- DTCs. The most frequently occurring profiles among CK+ DTCs were the mixed phenotypes CK+Vim+Ki67+ and CK+Vim+Ki67+PDL1+. Remarkably, the majority of solely epithelial cells (CK+/Vim-) were negative for PD-L1/Ki67/HER2. We observed an association of Ki67+ DTCs (in combination with PD-L1 or HER2) with increasing mesenchymal features (CK-/Vim+). Among the CK- DTCs, the profile Vim+Ki67+ was prepotent. Our data indicate that vimentin might play a role in tumor cell dissemination, which in turn is associated with recurrence and metastasis. Previous studies in lung- and colon cancer reported that increased vimentin expression was connected with metastasis, shorter survival, and affected lymph nodes [46,47]. In TNBC, the occurrence of Vim+ tumor cells was associated with bad prognosis [48,49]. Further, vimentin-expressing tumor cells were described to be poorly differentiated and showed properties linked to infiltration and malignancy [50]. In a TNBC patient-derived mouse model, Grasset et al. observed that metastasis involves EMT dynamics and included a high fraction of hybrid epithelial/mesenchymal cells, which led to invasion in their models and patients. They reported that vimentin is required for the formation of metastasis in 3D cultures and mouse models. Further, in the patient metastasis, vimentin was expressed suggesting that the mesenchymal character of TNBC was retained [49].

Here, our study revealed the occurrence of DTCs with mesenchymal features (CK-/Vim+) in patients that were considered DTC negative according to the CK based standardized detection method. For example, one patient (ID 1230) was diagnosed with TNBC in 07/2020 and no CK+ DTCs were detected. However, the subsequent multi-parameter staining discovered CK-/Vim+ DTCs. In discordance to the primary tumor, she presented HER2+ DTCs. Concordantly, she had PD-L1+ and Ki67+ DTCs, while the primary tumor was PD-L1 positive with a tumor proportion score (TPS) of 2 and combined positive score (CPS) of 4. The Ki67 index was 40% in the CNB and 75% in the resected tumor tissue after NACT without reaching pCR. Noteworthy, in 03/22 the patient was diagnosed with recurrent disease. Another patient (ID 1085) was diagnosed with TNBC in 04/2022 in addition to a second malignancy of the ovaries. No CK+ DTCs were detected using the standardized bright field method. However, the novel staining approach revealed HER2+ and PD-L1+ as well as Ki67+ DTCs, while the primary tumor was HER2 negative and PD-L1 negative. The Ki67 index was 70% in the CNB. The patient reached pCR after NACT but presented with pulmonary metastasis in 09/2023. One patient (ID 3217) was diagnosed with TNBC in both breasts in 07/2022 and CK+ DTCs. She reached pCR after NACT and oral bisphosphonate intake was applied for 24 months after surgery. However, she was diagnosed with HER2+ recurrent disease in the left breast in 03/2024. Interestingly, the sequential staining procedure revealed that the CK+ DTCs were also Vim+, PD-L1+, Ki67+, and HER2+. Another case concerning HER2+ DTCs (ID 3306) was diagnosed with TNBC in 09/2022 and the resected tumor tissue turned out to be HER2 low (Score 1+) and PD-L1 positive (TPS 1, CPS 4) with an Ki67 index of 80%, while pCR was reached and pembrolizumab was administered. Using the standardized method, CK+ DTCs were detected and bisphosphonates were prescribed. However, the patient suffered from early local recurrence in 06/2023 with lymphangiosis carcinomatosa and died in 11/2023. The sequential staining method revealed that the CK+ DTCs were also positive for Vim, PD-L1, Ki67, and HER2 (Table A1). Hence, our study underlined the possibly prognostic relevance of Vim+, PD-L1+, and HER2+ DTCs.

A few studies already discovered HER2 expressing DTCs in early breast cancer patients that were initially diagnosed with HER2 negative primary tumors [51,52,53]. Our results are in line with research by Volmer et al. 2024 who emphasized the importance of HER2 expression on DTCs and subsequent potential therapeutic options [41]. Their study revealed that patients with initially HER2 negative tumors but HER2 positive DTCs had a worse outcome. In gastric cancer, the treatment with trastuzumab was already shown to be effective when patients with HER2 negative tumors had HER2 positive circulating tumor cells (CTCs) in the blood [54]. In metastatic breast cancer, the DETECT III trial revealed that patients with initially HER2 negative tumors but HER2 positive CTCs could benefit from additional treatment with lapatinib, a HER 2 tyrosine kinase inhibitor [55].

Another noteworthy aspect is the increased number of CK+ DTCs in pre-menopausal patients compared to peri- and post-menopausal cases (4.4 vs. 3.8 vs. 3.9). Although the patient numbers were too little for statistically significant results, this finding draws attention to the question whether or not menopausal status should be a decision criterion for treatment with bone targeting agents such as bisphosphonates or denusomab. Currently available data imply a benefit of bisphosphonate intake for post-menopausal patients only [34,56,57].

However, several limitations of the study must be considered. Since TNBC occurs in about 20% of breast cancer patients, the size of our cohort is relatively small (*n* = 80) and leads to a limited generalizability. Further, bone marrow aspirates were only collected at a single point of time, namely, during surgery. Thus, there is no insight into potential DTC dynamics, especially before and after NACT. A comparative evaluation of pre- and post-treatment samples, as applied by Wimberger et al. [45], could provide conclusions concerning the persistence of DTCs and their potential phenotypic alterations induced by NACT.

Since DTCs are rare events in bone marrow, their detection and characterization remain challenging. Technical limitations of immunofluorescence include potential spectral overlap, autofluorescence of bone marrow and variability in antibody sensitivity. Moreover, the image evaluation by a single observer may lead to potential bias and subjectivity.

Future investigations should validate these findings in larger, multi-center cohorts and consider DTC dynamics over time. A staining of stem cell markers, such as ALDH and CD133 as well as EMT-related markers like TWIST1 and AKT2 could be interesting in further DTC subgroup analyses.

## 5. Conclusions

Our study delivers insights into phenotypic heterogeneity and distinct subpopulations of DTCs in patients with TNBC. Correlating DTC characteristics with clinical data enhances our understanding of the prognostic relevance of specific DTC subtypes, particularly in relation to achieving pCR. Our results strengthen the assumption that DTCs are potential biomarkers for disease progression. In addition, specific markers on DTCs might be used as therapeutic targets in the future.

## Figures and Tables

**Figure 1 cells-14-00857-f001:**
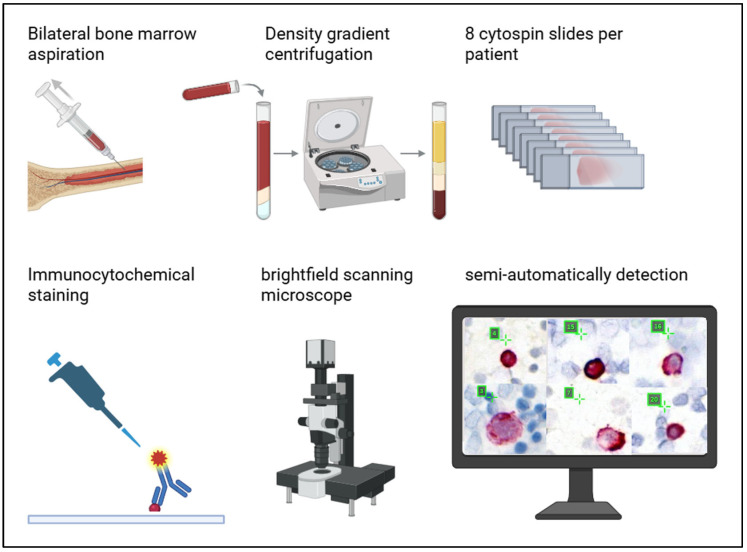
Standardized detection and staining of disseminated tumor cells (DTCs). During oncologic surgery bone marrow aspirates were collected from the anterior iliac crest. The samples were separated into distinct fractions using density gradient centrifugation. The interphase containing hematopoietic cells and DTCs were applied to glass slides and immunocytochemical staining was performed with antibodies against pan-cytokeratin. As a result of the labeling with alkaline phosphatase, DTCs were visualized in pink. Cell nuclei were stained light blue using hematoxylin. With our algorithm trained for rare events [30,35], we were able to detect DTCs candidates. Figure was created by bioRender.com.

**Figure 2 cells-14-00857-f002:**
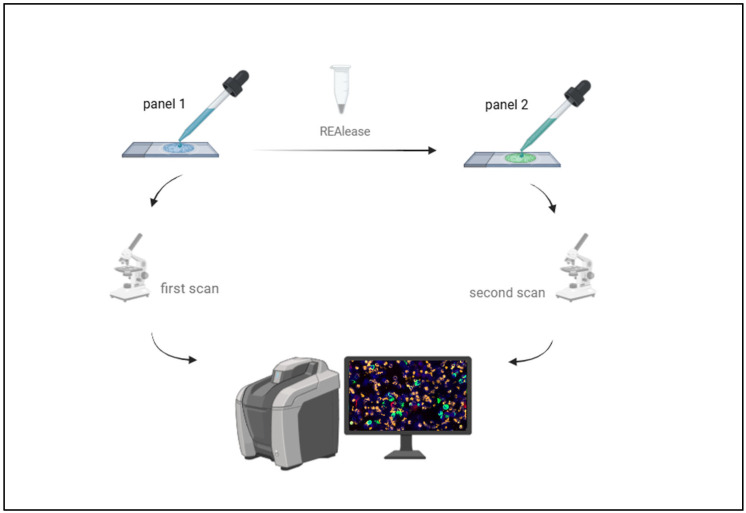
Schematic of sequential immunofluorescence staining. A two-staged staining process was performed with two different antibody panels and releasable fluorochrome-conjugates. After each staining, the slides were scanned with an immunofluorescence scanning microscope. Figure was created by bioRender.com.

**Figure 3 cells-14-00857-f003:**
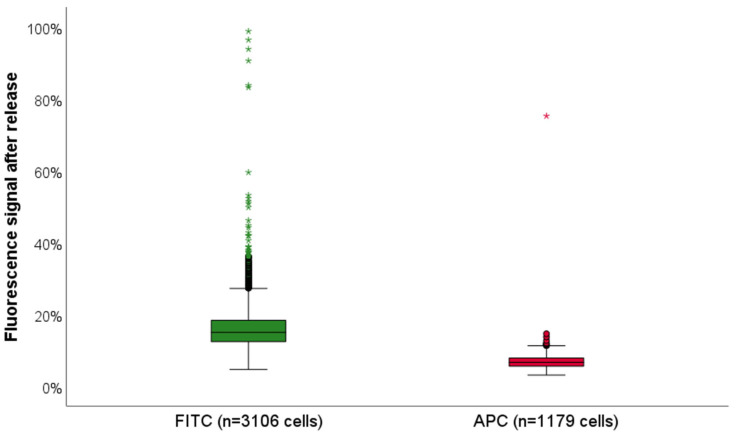
Quenching of the fluorescence signal after release. Fluorescence signal in the green (FITC; displayed in green) and red (APC, displayed in red) channel was reduced to a mean of 14.9%, respective 6.6%, after release. Asterisks represent extreme outliers among signal measurements in the FITC (green) and APC (red) channel. The release rate was only relevant in the red and green channel, as yellow (PE) was used for CD45 which was supposed to remain on the hematopoietic cells during further staining rounds and the white channel was used in panel 2; hence, there was no release required. Release rates were 85.1% in the green and 93.4% in the red channel.

**Figure 4 cells-14-00857-f004:**
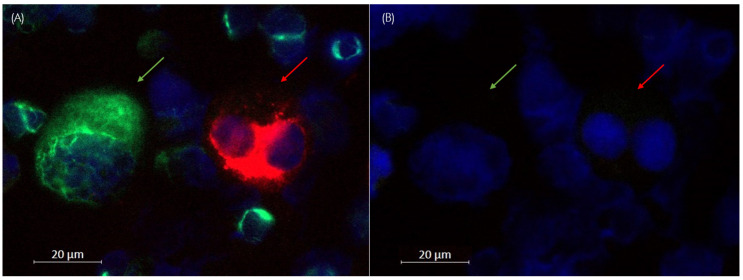
Pre- and post-release: Overlay of red, green, and blue channels showing immunofluorescent staining of a binucleated ZR75-1 cell (red arrows) and T98G cell (green arrows) against Pan-Cytokeratin (APC, red) and vimentin (FITC, green) mixed with bone marrow cells at 20-fold magnification. Nuclei were stained with DAPI (blue). (**A**) Before and (**B**) after treatment with the releasing enzyme.

**Figure 5 cells-14-00857-f005:**
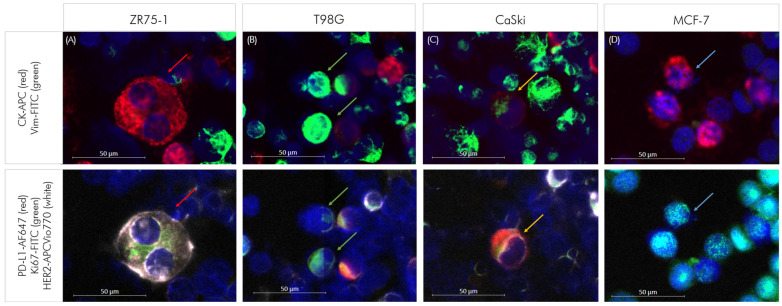
(**A**) ZR75-1 (red arrows): positive for cytokeratin (CK), HER2 and Ki67; (**B**) T98G (green arrows): positive for vimentin (Vim) and Ki67; (**C**) CaSki (yellow arrows): positive for CK, Vim, PD-L1, and Ki67 (**D**) MCF-7 (blue arrows): positive for CK and Ki67.

**Figure 6 cells-14-00857-f006:**
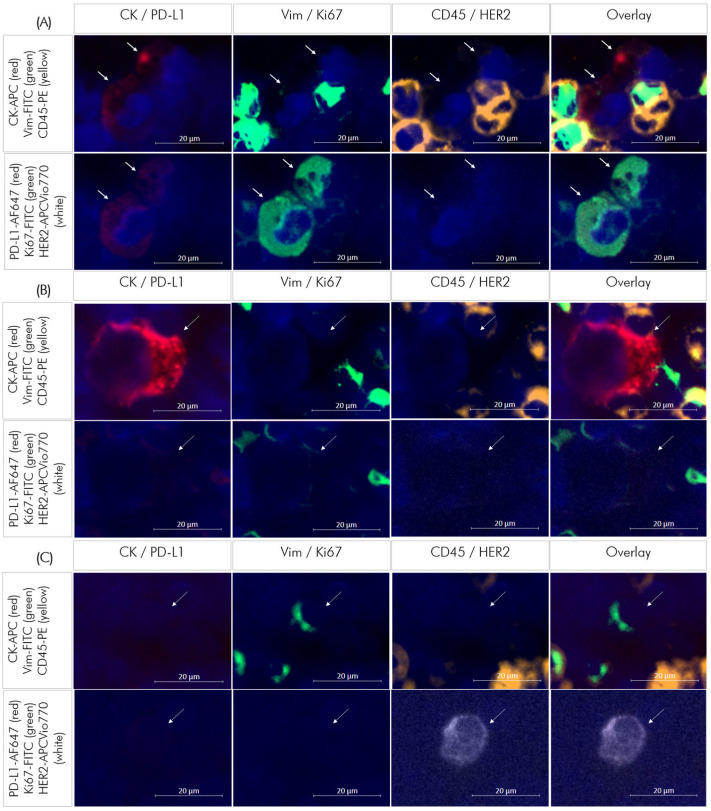
Representative images of disseminated tumor cells (DTCs) detected in three different patient samples. (**A**) DTC positive for cytokeratin (CK, red), PD-L1 (red) and Ki67 (green), negative for vimentin (Vim), CD45 and HER2. (**B**) CK positive (red) DTC, negative for all other markers. (**C**) HER2 positive (white) DTC, negative for all other markers.

**Figure 7 cells-14-00857-f007:**
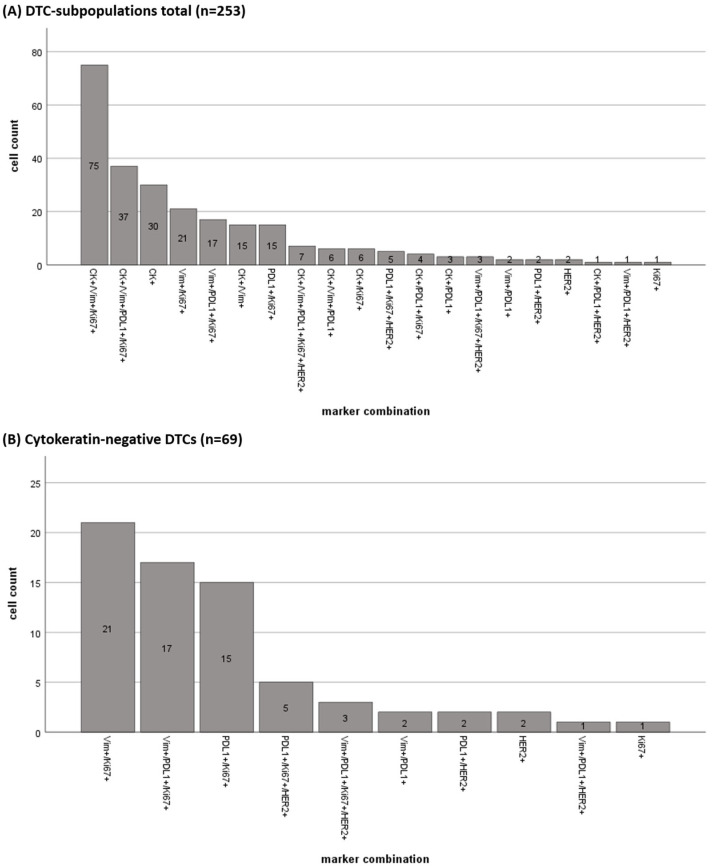
Quantification of disseminated tumor cells (DTCs) in (**A**) all subpopulations and (**B**) the Cytokeratin-negative DTC subpopulations.

**Figure 8 cells-14-00857-f008:**
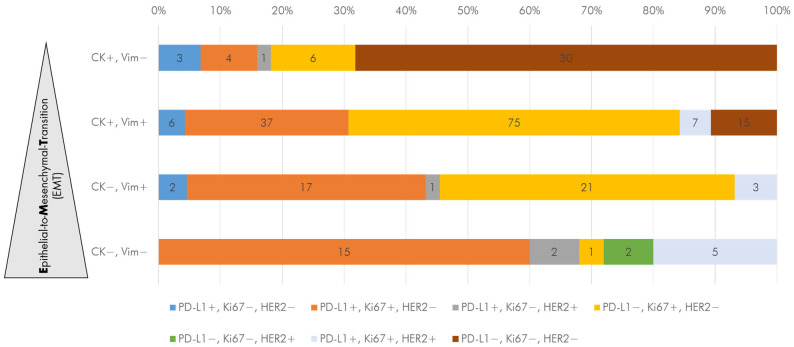
Considering the phenotype, we observed an association of Ki67+ DTCs (yellow bar) in combination with PD-L1 (orange) or HER2 (light blue) with increasing mesenchymal features (CK-/Vim+). Vice versa, the majority of DTCs with epithelial features (CK+, *n* = 30) was negative for PD-L1/Ki67/HER2 (brown bar). DTCs: disseminated tumor cells, CK: cytokeratin, Vim: Vimentin.

**Figure 9 cells-14-00857-f009:**
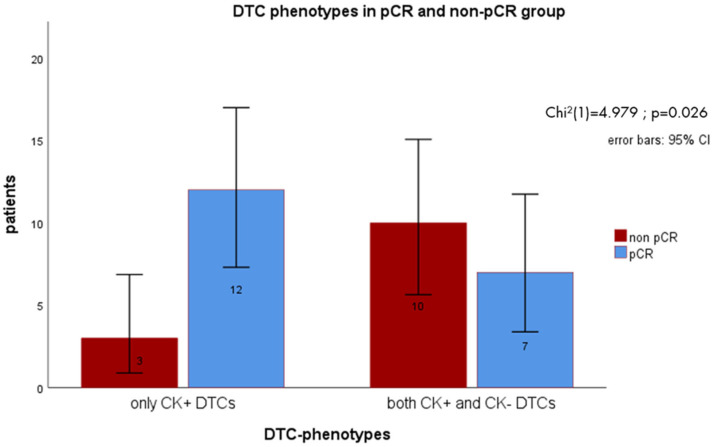
Disseminated tumor cells (DTCs) phenotypes in pCR (pathologic complete remission) and non-pCR group. DTC positive patients that received neoadjuvant chemotherapy (NACT) and reached pCR more frequently had only purely epithelial DTCs. Patients in the non-pCR group were more likely to have both cytokeratin-positive (CK+) and cytokeratin-negative (CK-) DTCs.

**Table 1 cells-14-00857-t001:** Patient characteristics.

	*n* = 80	%
**Age**
Median (years)	50	
Range	26–86	
**Menopausal status**
pre-	34	42.5
peri-	7	8.8
post-	39	48.8
**Tumor stage**
Primary	67	83.8
Recurrent	13	16.3
**Ki67 (CNB)**		
average (%)	52	
range (%)	2–90	
**Nodal status** N0 N1	59 21	73.8 26.2
**HER2 score (CNB)**		
0	64	80.0
1+	12	15.0
2+/CISH neg	4	5.0
**Grade (CNB)**		
G1	3	3.8
G2	28	35.0
G3	47	58.8
n.a.	2	2.5
**NACT**		
yes	64	80.0
combined with pembrolizumab	9	14.1
no	16	20.0
**pCR**	39	60.9

CNB: core needle biopsy, Ki67: proliferation index, NACT: neoadjuvant chemotherapy (mainly epirubicin, paclitaxel/carboplatin or docetaxel/carboplatin) pCR: pathologic complete remission.

**Table 2 cells-14-00857-t002:** Reference cells.

Cells/Marker	ZR75-1	T98G	CaSki	MCF7	Hematopoietic Cells
Cytokeratin	+	-	+	+	-
Vimentin	-	+	+	-	+
CD45	-	-	-	-	+
PD-L1	-	-	+	-	both *
Ki67	+	+	+	+	-
HER2	+	-	-	-	-
DAPI	+	+	+	+	+

Immunofluorescent staining of cell lines ZR75-1, T98G, CaSki, and MCF-7 as well as hematopoietic bone marrow cells for the six different markers used. + means that cells were stained positive and - means that they were negative for the respective markers. * In this study, the majority of examined hematopoietic cells was negative for PD-L1 and a proportion of approximately 10% showed a positive signal for PD-L1.

**Table 3 cells-14-00857-t003:** Receptor concordances and discordances between disseminated tumor cells (DTCs) and tumor tissue.

Marker	Patients with DTCs, Positive for Marker (*n*)	Tumor Tissue * Positive (*n*)	Tumor Tissue * Negative (*n*)	Missing Data (*n*)
HER2	9	4	3	2
PD-L1	27	10	10	7

Of the nine patients with HER2 positive DTCs, we analyzed tumor tissue in seven cases and found concordance in four cases. In the cohort, 27 patients had PD-L1 positive DTCs, and concordance was found in 10 of 20 analyzed tumor tissues. * tumor tissue either from CNB, resected tumor tissue during surgery, or lymph node biopsy.

## Data Availability

The original contributions presented in this study are included in the article. Further inquiries can be directed to the corresponding author.

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
