# Peer review of "Characterization of Disseminated Tumor Cells (DTCs) in Patients with Triple-Negative Breast Cancer (TNBC)"

_cells, 2025, doi:10.3390/cells14120857_

Round 1
Reviewer 1 Report
Comments and Suggestions for Authors
Triple negative breast cancer (TNBC) is a more aggressive molecular subtype, so this study is interesting and may contribute to improving prognostic factors and therapy directly targeting, through the characterization of disseminated tumor cells (DTCs), despite its technical limitations.
The topic is relevant to the area, as it investigates the presence of disseminated tumor cells (DTCs) with specific markers and their potential prognostic value in TNBC. The originality is not in the topic, but in the approach adopted, which may contribute with additional data in the future.
The study adds to the evidence by demonstrating the presence of disseminated tumor cells (DTCs) in approximately 50% of the patients analyzed and their possible markers, relevant evidence although obtained with limited methodology. This finding reinforces the need to investigate the role of these cells in recurrence and resistance to treatment in TNBC.
The study is very clear in its results and discussion.
The conclusions of this study are consistent with the data presented and are well-founded in the literature. The main question of the study was adequately addressed, although with methodological limitations that should be considered.
The references used are appropriate.
Follow suggestions for authors:
What criteria have be used to characterize the DTCs as positive in the samples?
Of the 45 patients detected with DTCs in relation to age, were they the youngest?
Since TNBC is a more aggressive molecular subtype, is there a relationship between the presence of DTCs and survival?
The criteria used to define the positivity of the samples is not clear and requires further detailing.
The inclusion of negative controls would be important to exclude the possibility of cross-reaction.
The lack of comparative analysis between pre- and post-treatment samples limits the conclusions about the persistence of CTDs; such analysis could generate more robust data.
Validation of the findings in another in vitro model, especially in relation to treatment resistance, would strengthen the conclusions.
The use of new markers, already suggested in the discussion, could increase the robustness of the analysis. However, what would be the strategy to obtain sufficient material for the new markers?
Author Response
We thank reviewer 1 for the positive feedback and valuable suggestions.
What criteria have be used to characterize the DTCs as positive in the samples?
This is an essential point and we have described the matter in the results section point 3.2 Identification and Quantification of DTC-Subtypes in Bone Marrow Samples : “When examining bone marrow samples from breast cancer patients, cells that had an increased nucleus/plasma-ratio and showed a negative staining against CD45, positive nuclear staining with DAPI and a positive staining against Pan-CK, Vim, HER2, Ki67, PD-L1 or multiple markers at once were captured and considered as tumor cells.”
Of the 45 patients detected with DTCs in relation to age, were they the youngest?
Thank you for this interesting question. There was no correlation between age at diagnosis and DTC-status nor DTC count using spearman rho correlation. The median age of DTC positive patients was 52.9 years vs. 51.2 years in DTC negative patients.
Since TNBC is a more aggressive molecular subtype, is there a relationship between the presence of DTCs and survival?
We thank the reviewer for this very relevant question. At the time being we do not have complete follow up data from all patients included in this study. Usually patients attend to their local gynaecologists for after care and only return to our department in case of recurrence. As we analyse bone marrow not only from patients with TNBC, but from all molecular subtypes, we conducted a systemic follow up of the entire cohort which inclues >600 cases. Therefore, it will still take some time. However, the association between DTCs and survival has been shown manifold in the literature which we mentioned in the introduction:
Braun, S.; Vogl, F.D.; Naume, B.; Janni, W.; Osborne, M.P.; Coombes, R.C.; Schlimok, G.; Diel, I.J.; Gerber, B.; Gebauer, G.; et al. A pooled analysis of bone marrow micrometastasis in breast cancer. N. Engl. J. Med. 2005, 353, 793–802, 594. doi:10.1056/NEJMoa050434.
Hartkopf, A.D.; Brucker, S.Y.; Taran, F.-A.; Harbeck, N.; Au, A. von; Naume, B.; Pierga, J.-Y.; Hoffmann, O.; Beckmann, M.W.;Rydén, L.; et al. Disseminated tumour cells from the bone marrow of early breast cancer patients: Results from an international pooled analysis. European journal of cancer (Oxford, England : 1990) 2021, 154, 128–137, doi:10.1016/j.ejca.2021.06.028.
Janni, W.; Vogl, F.D.; Wiedswang, G.; Synnestvedt, M.; Fehm, T.; Jückstock, J.; Borgen, E.; Rack, B.; Braun, S.; Sommer, H.; et al. Persistence of disseminated tumor cells in the bone marrow of breast cancer patients predicts increased risk for relapse--a European pooled analysis. Clin. Cancer Res. 2011, 17, 2967–2976, doi:10.1158/1078-0432.CCR-10-2515.
Hartkopf, A.D.; Taran, F.-A.; Wallwiener, M.; Hahn, M.; Becker, S.; Solomayer, E.-F.; Brucker, S.Y.; Fehm, T.N.; Wallwiener, D. Prognostic relevance of disseminated tumour cells from the bone marrow of early stage breast cancer patients - results from a large single-centre analysis. European journal of cancer (Oxford, England : 1990) 2014, 50, 2550–2559, doi:10.1016/j.ejca.2014.06.025.
The criteria used to define the positivity of the samples is not clear and requires further detailing.
Thank you for ponting out that our description is not clear enough. For positivity of DTCs using the standard brightflied method, we added the following passage in the methodolog section 2.2 Bone Marrow Aspirates: In brief, DTCs were visualized in pink using alkaline phosphatase and short counterstaining with hematoxylin which colored the nuclei light blue. DTCs were semi-automatically detected and enumerated using the Aperio Versa microscope-based scanning system (Leica Biosystems) with rare events software that was trained to select DTC candidates according to color, shape, intensity and size. Two independent investigators, a certified cytologist and a trained pathologist, individually evaluated cell morphology and cytological staining patterns of the selected DTC candidates.
In the material & methods section 2.4. Sequential Immunofluorescence Staining and DTC Imaging, we added the following sentence: Cells that were CD45 negative, but had a positive nuclear staining with DAPI and a positive staining against Pan-CK, Vim, HER2, Ki67, PD-L1 or multiple markers at once were defined as tumor cells.
As mentioned above, the detection of DTCs using the seqeuntial immunofluorescent aproach is described as part of the results section 3.2. Identification and Quantification of DTC-Subtypes in Bone Marrow Samples, as this pattern was developed throughout the experimental set up: When examining bone marrow samples from breast cancer patients, cells that had an increased nucleus/plasma-ratio and showed a negative staining against CD45, positive nuclear staining with DAPI and a positive staining against Pan-CK, Vim, HER2, Ki67, PD-L1 or multiple markers at once were recorded and considered as tumor cells.
The inclusion of negative controls would be important to exclude the possibility of cross-reaction.
This is an essential point as well. As we are detecting rare cells among millions of bone marrow cells, we use these hematopoietic cells as internal negative controll on each slide. Further, we used a control slide with MCF7 cells as negative controls for vimentin, PD-L1 and HER2 with each run. We added this in the Material & Methods passage 2.4. Sequential Immunofluorescence Staining and DTC Imaging : As a positive and negative control with each run, we used reference slides with a mix of bone marrow cells and ZR75-1 (breast cancer cells), CaSki (cervical carcinoma cells) plus T98G (glioblastoma cells) as wells as slides with MCF-7 cells which were prepared, stored, and fixed in the same manner. Further, the marker espressions are shown in Tabel 2. Reference Cells.
The lack of comparative analysis between pre- and post-treatment samples limits the conclusions about the persistence of CTDs; such analysis could generate more robust data.
We absolutely agree with the reviewer. However, we have to comply with the ethical aspects of the study. Bone marrow aspiration is not a minimal-invasive sample collection approach. The procedure has to be done using at least local anesthesia, better during tumor surgery. Usually, TNBC patients receive neoadjuvant chemotherapy. That means, bone marow should be sampled at diagnosis already and patients should give written consent when they were just confronted with many serious live changing facts. Not many patients would agree to (re-) punctuation, therefore, it would narrow the cohort down to very small numbers. Further, the aim of the study was not to show DTC eradication, but to characterize DTCs. We added a sentence in the discussion addressing that point: “A comparative evaluation of pre- and post-treatment samples as applied by Wimberger et al. [46] could provide conclu-sions concerning the persistence of DTCs and their potential phenotypic alterations inducted by NACT.”
Validation of the findings in another in vitro model, especially in relation to treatment resistance, would strengthen the conclusions.
We thank the reviewer for this interesing thought. As DTCs are heterogenous and rare events, it is a huge challenge to isolate and culture/treat them. Currently, we are testing a marker-free system to isolate DTCs from cryo-preserved bone marrow samples. In the case of success, we might conduct cell culture experiments testing the response rate of DTCs to various treatments based on the markers they express.
The use of new markers, already suggested in the discussion, could increase the robustness of the analysis. However, what would be the strategy to obtain sufficient material for the new markers?
This is a really good point. To visualize a greater number of markers a method is required that enables either parallel or sequential staining on the same sample. This could be accomplished by using multiple filter channels on the fluorescence microscope. Alternatively, we described a bleaching-based approach in a previous study on hormone-receptor-positive breast cancer patients that allows additional sequential staining steps by quenching (bleaching) fluorochromes:
König, Theresa; Dogan, Senol; Höhn, Anne Kathrin; Weydandt, Laura; Aktas, Bahriye; Nel, Ivonne (2023): Multi-Parameter Analysis of Disseminated Tumor Cells (DTCs) in Early Breast Cancer Patients with Hormone-Receptor-Positive Tumors. In: Cancers 15 (3). DOI: 10.3390/cancers15030568.
Reviewer 2 Report
Comments and Suggestions for Authors
The article is valuable in highlighting the need for greater understanding of the nature of disseminated tumor cells in TNBC, which is still a major medical problem affecting significant proportion of patients with mammalian cancers. I consider the strengths of the manuscript to be a thorough introduction, description of the methodology, and discussion.
The authors should explain why there is significantly more variance in the values for quenching of the fluorescence signal in green (FITC) channel as compared to red /APC) channel, and whether this may affected the results (Figure 3).
Table 2, Characterization of reference cell lines and cells: Were the hematopoietic cells really negative for CD45? The presence of CD45 on hematopoietic cells is considered one of their essential markers. CD45 negativity would call into question the whole system of reference cells used in the study.
In the description of the methods I would point out some minor points:
The source of the tumor cell lines used as controls and the hematopoietic cells are missing. It is a common indication. Paragraph 2.3, line 152 onwards.
Further points of formal character:
There is no need to explain the basic terms in the figure legends repeatedly, even though only the abbreviation is used throughout the text. E.g., "disseminated tumor cells, DTC, in the legends of Fig. 1, 6, 7, 8, 9, Table 3. Similarly, vimentin, pathologic complete remission, etc. If this is the editor's request, then yes. However, I find it unnecessary, little used, and "slowing down" the reading. Moreover, a list of abbreviations is also provided.
The legend to Figure 3 is also embedded in the text (lines 267 - 272).
Line 115: summarized instead of summerized
Signs + and - in the sense of "positive" and "negative" for individual markers or populations (CK+, DTC+ etc. should be written as superscript (CD+, DTC+) - occurs many times in the text.
Table 2 would be much clearer when using just "+" and "-" instead of "positive" and "negative"
Author Response
We thank reviewer 2 for the positive and thorough critique.
The authors should explain why there is significantly more variance in the values for quenching of the fluorescence signal in green (FITC) channel as compared to red /APC) channel, and whether this may affected the results (Figure 3).
Thank you for this rather technical question. The variation in quenching values in the green channel compared to the red channel is primarily due to the increased autofluorescence of the bone marrow, which is most prominent in the green channel. Since this autofluorescence persists after the release step, the quenching efficiency in the green channel seems lower than in the red channel. However, this background signal is also visible in the negative controlls and can therefore be accounted during analysis. Additionally, there is a distinct distribution pattern of marker expression in the green channel between Vimentin in panel 1 and Ki67 in panel 2, which further supports the validity of our evaluation. We added a snetence adressing this point in the results 3.1.2 Calculation of the release rate: “Due to persisting autofluorescence in the bone marrow samples, signal reduction was lower in the green compared to the red channel. However, the background signal was also detetected in the negative controls and accounted accordingly during analysis.”…” In addition, the markers vimentin (used in the first panel) and Ki67 (used in the second panel) labelled with the green dye, showed different expression patterns.”
Table 2, Characterization of reference cell lines and cells: Were the hematopoietic cells really negative for CD45? The presence of CD45 on hematopoietic cells is considered one of their essential markers. CD45 negativity would call into question the whole system of reference cells used in the study.
We thank the reviewer for the through review and appreciate pointing out this essential mistake in Table 2. As we described in the main text, we used CD45 as exclusion marker, so of course the hematopoietic cells were positive for CD45. We changed tabel 2 accordingly.
The source of the tumor cell lines used as controls and the hematopoietic cells are missing. It is a common indication. Paragraph 2.3, line 152 onwards.
Point well taken, we added the info accordingly. The cells were obtained from the ATCC, multiplied and cryo-preserved.
There is no need to explain the basic terms in the figure legends repeatedly, even though only the abbreviation is used throughout the text. E.g., "disseminated tumor cells, DTC, in the legends of Fig. 1, 6, 7, 8, 9, Table 3. Similarly, vimentin, pathologic complete remission, etc. If this is the editor's request, then yes. However, I find it unnecessary, little used, and "slowing down" the reading. Moreover, a list of abbreviations is also provided.
We really agree with the reviewer, however, after submitting the manuscript, the editor specifically asked us to revise the figure legends in order to understand them without reading the main text: “providing the full names for abbreviations used.”
The legend to Figure 3 is also embedded in the text (lines 267 - 272).
See comment by editor, cited above.
Line 115: summarized instead of summerized
We thank the reviewer for the careful proof-read and corrected the typo.
Signs + and - in the sense of "positive" and "negative" for individual markers or populations (CK+, DTC+ etc. should be written as superscript (CD+, DTC+) - occurs many times in the text.
This is a good idea, however, we found superscript difficult to read and after careful consideration and internal discussions decided to keep it + and – in the manuscript.
Table 2 would be much clearer when using just "+" and "-" instead of "positive" and "negative"
This is a valuable advice. We changed Table 2 accordingly. Thank you.
Round 2
Reviewer 1 Report
Comments and Suggestions for Authors
The authors have addressed the requests, adequately clarifying the issues raised. The introduction and discussion sections have been revised to enhance the understanding of the topic. Additionally, further details regarding the technique used and the analysis performed have been included, which have been incorporated into the methodology and discussed in the results. It is emphasized that no new cases were added, nor were there any changes to the previously presented methodology. The justifications provided were appropriately supported by relevant references in each item.
Author Response
We thank Reviewer 1 for the positive comments.